

# Epigenetic regulation of dental-derived stem cells and their application in pulp and periodontal regeneration

Yuyang Chen[1,*], Xiayi Wang[1,*], Zhuoxuan Wu[1], Shiyu Jia[1] and Mian Wan[1,2]

[1] State Key Laboratory of Oral Diseases & National Clinical Research Center for Oral Diseases & West China School of Stomatology, Sichuan University, Chengdu, Sichuan, People's Republic of China
[2] State Key Laboratory of Oral Diseases & National Clinical Research Center for Oral Diseases & Department of Cariology and Endodontics, West China Hospital of Stomatology, Sichuan University, Chengdu, Sichuan, People's Republic of China
* These authors contributed equally to this work.

Corresponding author
Mian Wan, mianwan@scu.edu.cn

## ABSTRACT

Dental-derived stem cells have excellent proliferation ability and multi-directional differentiation potential, making them an important research target in tissue engineering. An increasing number of dental-derived stem cells have been discovered recently, including dental pulp stem cells (DPSCs), stem cells from exfoliated deciduous teeth (SHEDs), stem cells from apical papilla (SCAPs), dental follicle precursor cells (DFPCs), and periodontal ligament stem cells (PDLSCs). These stem cells have significant application prospects in tissue regeneration because they are found in an abundance of sources, and they have good biocompatibility and are highly effective. The biological functions of dental-derived stem cells are regulated in many ways. Epigenetic regulation means changing the expression level and function of a gene without changing its sequence. Epigenetic regulation is involved in many biological processes, such as embryonic development, bone homeostasis, and the fate of stem cells. Existing studies have shown that dental-derived stem cells are also regulated by epigenetic modifications. Pulp and periodontal regeneration refers to the practice of replacing damaged pulp and periodontal tissue and restoring the tissue structure and function under normal physiological conditions. This treatment has better therapeutic effects than traditional treatments. This article reviews the recent research on the mechanism of epigenetic regulation of dental-derived stem cells, and the core issues surrounding the practical application and future use of pulp and periodontal regeneration.

## INTRODUCTION

Dental-derived stem cells are obtained from dental papilla or dental follicles and can be isolated and cultured in teeth or periodontal soft tissue. Dental pulp stem cells (DPSCs), stem cells from exfoliated deciduous teeth (DFPCs), and stem cells from apical papilla (SCAPs) are all derived from dental papillae. Dental follicle precursor cells (DFPCs) and periodontal ligament stem cells (PDLSCs) are derived from dental follicles. In recent years, the discovery of dental-derived stem cells, their abundant sources, and their safety

and effectiveness have won them increasing attention in the field of tissue regeneration. Epigenetic regulation refers to the regulation of gene expression without changing the DNA sequence. This plays an important role in the self-renewal and differentiation capacity of adult and embryonic stem cells (*Chen et al., 2017*). Epigenetic regulation exists widely in natural organisms and participates in many biological processes, such as embryogenesis, germ cell formation, hematopoietic stem cell differentiation, and tumor formation (*Canovas et al., 2017*; *Chen, Yan & Duan, 2016*; *Mohammad, Barbash & Creasy, 2019*). More importantly, epigenetic modifications can not only regulate tooth formation, development, and aging, but also affect the differentiation of dental-derived stem cells (*Hodjat, Khan & Saadat, 2020*; *Liu et al., 2021*; *Townsend et al., 2009*). Epigenetic modifications are advantageous because they do not cause permanent DNA damage, off-target effects, or deleterious mutations. Therefore, an increasing number of studies have focused on the role of epigenetic modifications in regulating the proliferation and differentiation of dental-derived stem cells. This article focuses on the epigenetic regulation of dental-derived stem cells, describes the application of epigenetic regulation based on dental-derived stem cells in dental pulp and periodontal regeneration, summarizes the shortcomings of the existing research, and proposes possible future research directions.

### Why this review is needed and who it is intended for

Endodontic and periodontal diseases are common and frequently-occurring diseases of the oral cavity. Traditional treatments for endodontics include root canal therapy and apical surgery. These treatments provide good relief of symptoms but cannot avoid tooth discoloration and pulp inactivation and necrosis. Periodontitis can cause destruction of the alveolar bone and also tooth loss. Traditional periodontal treatment may meet the needs of most patients with periodontitis, but fail to achieve periodontal tissue regeneration. In recent years, with the development of tissue engineering and the discovery of dental-derived stem cells, pulp and periodontal regeneration have become a potential treatment for these two diseases. The discovery of the mechanism of epigenetic modification of dental-derived stem cells has led to the potential use of epigenetic regulation in dental pulp and periodontal regeneration. However, no article targeting the same subject has been published to provide a clear guidance for clinic trials.

This article aims to provide perspectives for dentists and researchers investigating dental pulp and periodontal regeneration using epigenetic regulated dental-derived stem cells, in order to reduce failures and improve the prognosis for patients. It is our hope that these techniques can develop more secure and effective treatment approaches.

## SURVEY METHODOLOGY

To ensure an inclusive and unbiased analysis of the literature and to accomplish the review's objectives, we searched the following literature databases: PubMed, Science Direct, Research Gate, and Google Scholar. The search terms included: dental-derived stem cells, epigenetic regulation, dental pulp regeneration, periodontal regeneration, DPSCs, SHEDs, SCAPs, DFPCs, PDLSCs, the targets were searched together with Boolean operators such as ''AND'' and ''OR''. It is important to note that the keywords used and their variants and relevant

words could be classified into categories and any combination of words from different categories was used for the search. The categories we used were as follows:

1. About dental-derived stem cells: dental-derived stem cells; DPSCs; SHEDs; SCAPs; DFPCs; PDLSCs; odontogenic stem cells.

2. About epigenetic regulation: epigenetic regulation; epigenetics; DNA methylation; histone modifications; noncoding RNA.

3. About pulp and periodontal regeneration: pulp regeneration; periodontal regeneration; odontogenic differentiation; nerve regeneration; angiogenesis; osteogenic differentiation.

This article is based on published literature. The aspects of the inclusion criteria are the retrieval keywords, information not covered by previous literature, and most importantly the use of a clear and credible source.

## Properties of dental-derived stem cells

Stem cells have significant proliferative capacity and multi-directional differentiation potential. The development of regenerative medicine has led to the use of stem cells in the repair of damaged cells, tissues, and organs, which have low self-healing abilities, with excellent safety and efficacy. Dental-derived stem cells mediate the process of tooth regeneration by upregulating odontogenic and angiogenic capacity in the form of secreted exosomes (Exo) (*Mai et al., 2021*). Dental-derived stem cells have been used in relevant clinical trials in the fields of periodontal tissue, maxillofacial bone tissue repair, and apical pulp disease treatment (*Feng et al., 2010*; *Giuliani et al., 2013*; *Nakashima & Iohara, 2014*). However, the preclinical models for the use of stem cells in nerve regeneration, diabetes, and autoimmune diseases have only been preliminarily validated (*Kanafi et al., 2013*; *Mead et al., 2017*; *Shimojima et al., 2016*).

## Properties and potential of DPSCs

Dental pulp stem cells have a relatively high proliferation rate, a low cellular senescence, multi-directional differentiation potential, and immunomodulatory properties (*Ma et al., 2019*; *Mortada & Mortada, 2018*). Damage to dental pulp causes dental pulp stem cells to induce the formation of various cellular components, including odontoblasts, to replenish damaged cells. In addition, *in vitro* studies have shown that DPSCs can differentiate into neural-like cells, osteoblasts, chondrocytes, adipocytes, muscle cells, endothelial cells, hepatocytes, and renal pericytes, *etc.* (*Barros et al., 2015*; *Gandia et al., 2008*; *Saito et al., 2015*). DPSCs can effectively promote pulp and periodontal regeneration. *Guo et al. (2021)* combined decellularized tooth matrix (DTM) with human dental pulp stem cells and successfully achieved the regeneration of dental pulp and periodontal tissue.

## Properties and potential of *SHEDs*

SHEDs are isolated from the pulp tissue of exfoliated deciduous teeth, whose expression level of osteocalcin and alkaline phosphatase activity are higher compared to DPSCs (*Koyama et al., 2009*). SHEDs can express osteocalcin and RUNX-2 markers, resulting in the differentiation potential of osteoblasts and odontoblasts (*Miura et al., 2003*). SHEDs can induce the migration of naive bone marrow mesenchymal stromal cells (BMSCs) by

secreting extracellular vesicles (EVs) containing various cytokines, thereby promoting the bone healing process (*Luo, Avery & Waddington, 2021*). SHEDs can also induce pulp and periodontal regeneration. *Yang et al. (2019)* combined SHEDs cell sheets and DFSCs cell sheets with dentin matrix (TDM) and implanted them into the orthotopic jawbone of nude mice. The results indicated that SHED/TDM successfully achieved periodontal tissue regeneration with better migration ability and neurogenic differentiation potential (*Yang et al., 2019*).

## Properties and potential of SCAPs

SCAPs exist in the apex of the developing tooth before tooth eruption and differentiate to odontoblasts, which mainly secrete apical dentin. SCAPs are less resistant to immune cell-mediated toxicity compared with other dental-derived stem cells, but can induce high levels of pro-inflammatory cytokine secretion (*Whiting et al., 2018*). Since SCAPs are derived from developing odontogenic tissues, they are more widely used in the field of tissue regenerative. For example, in dental pulp engineering, the regeneration process of dental pulp can be achieved by inducing endogenous stem cells to move to the regeneration site (*Rombouts et al., 2017*). *Wei, Sun & Hou (2021)* successfully used the silk fibroin-RGD-stem cell factor scaffold (the RGD peptide was arginine-glycine-aspartic acid polypeptide) to promote the migration and proliferation of SCAPs. This approach is promising for the further use of cell homing in dental pulp regeneration.

## Properties and potential of DFPCs

The dental follicle contains a large number of undifferentiated precursor cells. In 2005, DFPCs were first isolated from the dental follicles of human third molars (*Morszeck et al., 2005*). DFPCs are derived from neural crest and are direct precursor cells of periodontal tissue (*Zhang et al., 2019a*). DFPCs can promote pulp regeneration through the paracrine pathway. *Hong et al. (2020)* found that DFPCs could effectively enhance the proliferation, migration, and odontogenic differentiation of inflammatory dental follicle cells (DPCs) *in vitro* and their ectopic dentinogenesis *in vivo*.

## Properties and potential of PDLSCs

Periodontitis often leads to the destruction of periodontal tissue and may even lead to tooth loss. After root canal treatment, the periodontal tissue is the only source of nutrition for the root canals. Multiple preclinical studies have demonstrated the effectiveness of PDLSCs to restore damaged periodontal tissue in periodontal regenerative therapy (*Li et al., 2020a*). PDLSCs are mainly isolated from the periodontal tissue of permanent teeth and can be further differentiated into osteoblasts, chondrocytes, and adipocytes under appropriate conditions (*Deng et al., 2018a*). Studies have shown that PDLSCs have the strongest osteogenic ability, followed by DPSCs, and DFPCs (*Qu et al., 2021*).

## Epigenetic regulations of dental-derived stem cells

Epigenetics refers to changing the expression level and function of a gene without changing its sequence. Its regulatory processes are regulated by signaling molecules whose interactions with neighboring cells induce appropriate transcriptional and epigenetic responses
(*Surani, Hayashi & Hajkova, 2007*). The way in which epigenetic mechanisms regulate gene expression related to environmental factors plays an important role in the development of various diseases, such as tumors and inflammation (*Yuan, Dong & Shen, 2022*; *Zarzour, Kim & Weintraub, 2019*). In addition, epigenetic regulation may also affect the tooth number, size, and shape (*Fernández et al., 2020*). The common epigenetic regulations include DNA methylation, histone modification, and non-coding RNA regulations. Tables 1–3 summarize the regulatory mechanisms and potential applications of DNA methylation, histone modifications, and ncRNAs in dental-derived stem cells.

## DNA methylation

DNA methylation is one of the important epigenetic modifications and it is common in most eukaryotes (*Lin et al., 2018*). The formation of 5-methylcytosine (5mC) from cytosine-phosphorothioate-guanine (CpG) dinucleotides by DNA methyltransferase (DNMT) leads to the silencing of gene expression (*Radhakrishnan, Kabekkodu & Satyamoorthy, 2011*). The level of methylated CpG is regulated by DNMT and DNA demethylase ten-eleven translocation (TET) (*Ren, Gao & Song, 2018*; *Li et al., 2015*).

DNA methylation levels are associated with stemness and the differentiation potential of dental-derived stem cells. For instance, DPSCs exhibits low DNA methylation levels and repressive mark H3K9Me2 enrichment, which is mediated by increased DNMT3B and G9a expression, respectively. This leads to decreased AKT phosphorylation and promotes osteogenesis (*Shen et al., 2019*). The decreased expression of the serine metabolism-related enzyme phosphoserine aminotransferase 1 (PSAT1) provides less methyl donor S-adenosylmethionine (SAM) for the methylation of the aging marker p16 (CDNK2A), resulting in the reduced stemness and osteogenic differentiation capacity of DPSCs (*Yang et al., 2021*). Wnt can effectively regulate the epigenetic mechanism of DPSCs. The short-term activation of Wnt signaling by Wnt-3A can cause a decrease in the content of 5-methylcytosine (5-mC) in DPSCs, which reduces the ability of DPSCs to differentiate into osteoblasts (*Uribe-Etxebarria et al., 2020*).

Different odontogenic stem cell genes have varied methylation levels and differentiation potentials. In DPSCs, PDLSCs, and DFPCs, the methylation of genes CD109 and SMAD3 are significantly different. At the transcriptional level, PDLSCs showed significantly higher expression levels of CD109, SMAD3, ALP, and RUNX2, which were identical to the differences in their DNA methylation profiles. The transcription levels of osteogenic differentiation-related factors and their osteogenic differentiation potential are higher in PDLSCs (*Ai et al., 2018*). The osteogenic differentiation process can be altered by modulating the methylation levels of specific genes in PDLSCs. Advanced glycation end-products (AGE) can increase the expression of DNMT1 and inhibit the methylation activation of calcitonin-related polypeptide $\alpha$ (CALCA) promoter, which inhibits the osteogenic differentiation of PDLSCs (*Wang et al., 2022*). Additionally, periostin (POSTN) can reduce the level of AGE receptors and DNA methylation of the CALCA promoter, thereby attenuating the inhibitory effect of AGE induction (*Wang et al., 2022*).

**Table 1   Regulation of DNA methylation in dental-derived stem cells.**

| Modification | Stem cell | Locus | Pathway mechanism | Target protein | Potential applications | Ref |
|---|---|---|---|---|---|---|
| DNA methylation | DPSCs | p16 | PSAT1 provides reduced SAM and decreased p16 methylation | PSAT1, PHGDH | Improve the pulp regeneration potential of aging DPSCs | *Yang et al. (2021)* |
| DNA demethylation | DPSCs | Wnt | WNT-3A activates Wnt signaling by diminishing their 5mC content. | NNMT | Induce epigenetic remodeling and pulp regeneration potential of DPSCs | *Uribe-Etxebarria et al. (2020)* |
| DNA demethylation | DPSCs | OSX, DLX5, RUNX2 | 5-Aza-CdR induced the expression of OSX, DLX5 and RUNX2 by decreasing DNA methylation. | DSPP, DMP1 | Promote the odontogenic growth and differentiation of DPSCs | *Zhang et al. (2015)* |
| DNA methylation | DPSCs | KDM6B | Alcohol suppressed KDM6B through dysregulating DNA methylation | ALP, BMP2, BMP4, DLX2, OCN, OPN | Promote osteogenic and odontogenic growth of dental mesenchymal stem cells | *Hoang et al. (2016)* |
| DNA methylation | PDLSCs | MIR31HG | Mechanical force downregulates MIR31HG through DNA methylation. | IL-6 | Inhibit hPDLSCs proliferation | *Han et al. (2021)* |
| DNA demethylation | PDLSCs | CALAL | POSTN attenuated the AGE-induced CALAL methylation | RUNX2, OSX, OPEN, RANGE | inhibit the osteogenic differentiation of PDLSCs | *Wang et al. (2022)* |
| DNA methylation | PDLSCs | DKK-1 | Downregulation of Tet1 and Tet2 leads to hypermethylation of DKK-1 promoter, activating WNT pathway. | FasL | Promote immunomodulation of PDLSCs | *Yu et al. (2019)* |
| DNA methylation | PDLSCs | TNFR-1 | High-glucose upregulates TNFR-1 via CpG island hypomethylation. | TNFR-1 protein | Aggravate viability reduction in hPDLSCs | *Luo et al. (2020)* |
| DNA methylation | SHEDs | IGF2 | IGF2 was induced via DNA methylation and RXR/RAR pathways activation. | RUNX2, ALP, BGLAP, DLX5 | Promote osteogenic differentiation of SHED | *Fanganiello et al. (2015)* |
| DNA demethylation | DFSCs | HOXA2 | HOTAIRM1 induced HOXA2 via DNA hypomethylation | DSPP, DMP1 | Induce osteogenic differentiation of DFSCs | *Chen et al. (2020b)* |

Notes.

DPSCs, dental pulp stem cells; PDLSCs, periodontal ligament stem cells; SHEDs, stem cells from exfoliated deciduous teeth; DFSCs, dental follicle stem cells; PSAT1, phosphoserine aminotransferase 1; PHGDH, phosphoglycerate; 5mC, 5methyl-cytosine; NNMT, Nicotinamide-N-methyltransferase; 5-Aza-CdR, 5-Aza-20-de-oxycytidine kinase 1; TNFR1, tumor necrosis factor-alpha receptor-1; RXR/RAR, Retinoid X Receptor/ Retinoic Acid Receptor; DSPP, dentin sialophosphoprotein; DMP1, dentin matrix protein 1; KDM6B, lysine (K)-specific demethylase 6B; IL-6, interleukin- 6; ALP, alkaline phosphatase.

## Histone modifications

Histones are located in the nucleus of eukaryotic cells and can form nucleosomes, the basic structure of chromatin, when bound to DNA. Modifications of amino acid residues in histone tails can cause structural changes in histones, which provide sites that can be recognized by specific proteins (*Strahl & Allis, 2000*). The regulation of specific genes can

Chen et al. (2023), *PeerJ*, DOI 10.7717/peerj.14550

**Table 2  Regulation of histone modifications in dental-derived stem cells.**

| Modification | Stem cell | Locus | Pathway mechanism | Target protein | Potential applications | Ref |
|---|---|---|---|---|---|---|
| Histone deacetylation | DPSCs | P21 | IGFBP7 activated SIRT1, resulting in a deacetylation of H3K36ac and reduction of p21 transcription | SIRT1 deacetylase | Prevent DPSCs senescence and promote tissue regeneration | *Li et al. (2022)* |
| Histone acetylation | DPSCs | HAT/KAT8 | WNT-3A activates Wnt by induces HAT expression and increased H3AC. | ACLY | Reduce the ability of DPSCs to differentiate into osteoblasts | *Uribe-Etxebarria et al. (2020)* |
| Histone acetylation and methylation | DPSCs | WNT3A, DVL3 | Ferutinin regulates Wnt/$\beta$-catenin pathway by H3K9 acetylation and H3K4 trimethylation. | Osteocalcin, collagen 1A1 | Direct DPSCs towards the osteogenic lineage | *Rolph et al. (2020)* |
| Histone demethylation | DPSCs | BMP2, HOX | KDM6B catalyzes the demethylation of H3K27me3 and activates BMP2 and HOX | ALP, BMP2, BMP4, DLX2, OCN, OPN | Promote osteogenic and odontogenic growth of dental mesenchymal stem cells | *Hoang et al. (2016)* |
| Histone demethylation | DFSCs | Wnt | Downregulated MEG3/EZH2 activated Wnt/$\beta$-catenin signaling pathway via demethylation on H3K27 | $\beta$-catenin and Wnt5a protein | Promotes the osteogenesis of DPSCs and DFPCs | *Deng et al. (2018b)* |
| Histone methylation | DFSCs | PTH1R | CHD7 activates PTH/PTH1R signaling pathway and interaction with H3K4me | RUNX2, SP7, BGLAP, DLX5, BMP2, COL1A1 | Promote osteogenic differentiation of DFSCs | *Liu et al. (2020a)* |
| Histone methylation | DFSCs | SFRP1 | WAY-316606 inhibits SFRP1 via histone H3K4me3 and activates Wnt pathway | $\beta$-catenin, RUNX2, ALP, osteocalcin, collagen | Maintain the nonmineralized state of PDL Progenitors | *Gopinathan et al. (2019)* |

Chen et al. (2023), *PeerJ*, DOI 10.7717/peerj.14550

**Table 2** (*continued*)

| Modification | Stem cell | Locus | Pathway mechanism | Target protein | Potential applications | Ref |
|---|---|---|---|---|---|---|
| Histone methylation | PDLSCs | COL1A1, RUNX2, IL-1$\beta$, CCL5 | LPS downregulated COL1A1, COL3A1, RUNX2 by H3K4me3 and upregulated CCL5, DEFA4, IL-1$\beta$ gene expression by H3K27me3 | COL1A1, COL3A1, RUNX2, CCL5, DEFA4, IL-1$\beta$ | Regulate periodontal lineage differentiation and the coordination of the periodontal inflammatory response | *Francis et al. (2019)* |
| Histone methylation | PDLSCs | RUNX2, MSX2, DLX5 | The H3K4me3 active methyl mark globally switch to the H3K27me3 repressive mark under osteogenic induction conditions. | DSPP, DMP1 | Induce osteogenic differentiation of DFSCs | *Francis et al. (2020)* |
| Histone demethylation | PDLSCs | IGFBP5 | BCOR inhibits IGFBP5 through histone K27 methylation. | ALP | Promote the odontoblast differentiation, proliferation, migration and mineralization of PDLSCs | *Han et al. (2017)* |
| Histone methylation | SCAPs | p15$^{INK4B}$, p27$^{Kip1}$ | KDM2A increased H3K4 trimethylation at loci p15 and p27 | cyclin-CDK | Inhibit cell proliferation of SCAPs | *Gao et al. (2013)* |

**Notes.**

DPSCs, dental pulp stem cells; DFSCs, dental follicle stem cells; PDLSCs, periodontal ligament stem cells; SCAPs, stem cells from apical papilla; H3AC, acetylated-Histone 3; H3K4me3, he histone H3 methylated at lysine 4; H3K27me3, the histone H3 methylated at lysine 27; H3K9me3, the histone H3 methylated at lysine 9; ACLY, ATP-citrate lyase enzyme; BMP2, bone morphogenic protein 2; HDAC3, histone deacetylase 3; MEG3, maternally expressed 3; EZH2, the enhancer of zeste homolog 2; PTH1R, parathyroid hormone receptor-1; PCNA, Proliferating cell nuclear antigen; CHD7, Chromodomain helicase DNA-binding protein 7; $\alpha$-SMA, alpha-smooth muscle actin; TNNT2, cardiac muscle troponin T; ACTC1, cardiac muscle; IGFBP5, insulin-like growth factor binding protein 5; BCOR, BCL6 co-repressor; p15INK4B, cyclin-dependent kinase inhibitor 2B; p27Kip1, cyclin-dependent kinase inhibitor 1B.

**Table 3** The regulatory role of ncRNAs in dental-derived stem cells.

| Modification | Stem cell | Locus | Pathway mechanism | Target protein | Potential applications | Ref |
|---|---|---|---|---|---|---|
| miRNA | DPSCs | TLR-4 | LPS activates lipopolysaccharide/TLR-4 signaling pathway by downregulating miR-140-5p. | TLR-4 | Enhance differentiation of DPSCs and inhibit proliferation | *Sun et al. (2017a)* |
| miRNA | DPSCs | Rac1 | miR-224-5p targets the 3′-untranslated region of Rac1 gene and downregulates Rac1. | MAPK8, caspase-3, caspase-9, Fas ligand | Potect DPSCs from apoptosis | *Qiao et al. (2020)* |
| miRNA | DPSCs | TGFBR1 | miR-24-3p and LEF1-AS1 sponged to regulate TGFBR1 expression. | RUNX2, OSX, ALP | Promote osteogenic differentiation of DPSCs | *Wu, Lian & Sun (2020)* |
| miRNA | DPSCs | CAB39 | miR-34a-3p activates AMP-K/mTOR signaling pathway by downregulating CAB39 | AMPK, mTOR | Downregulate alleviates senescence in DPSCs | *Zhang et al. (2021)* |
| miRNA | DPSCs | Foxq1 | miR-320b mediated Foxq1 upregulation after calcium hydroxide stimulation. | cyclin E1, cyclin D1 | Pomote proliferation of DPSCs | *Tu et al. (2018)* |
| miRNA | DPSCs | TLR4 | lncRNA-Ankrd26 promotes migration and osteogenesis via regulating miR-150-TLR4 signaling in MSCs | OSX, ALP | Promote dental pulp repair | *Li & Ge (2022)* |
| miRNA | DFSCs | Runx2, ALP and SPARC | miR-204 negatively targets the gene of Runx2, ALP and SPARC. | Runx2, ALP and SPARC | Promote osteogenic induction in DFSCs | *Ito et al. (2020)* |
| miRNA | PDLSCs | IL-17, IL-35 | Overexpression of miRNA-146a downregulates IL-17 and IL-35 expression under periodontitis | IL-17, IL-35 | Inhibit proliferation of hPDLSCs | *Zhao, Cheng & Kim (2019)* |
| miRNA | PDLSCs | PTEN | miR-181b-5p regulates PTEN/AKT pathway and promotes BMP2/ Runx2 | PKB, BMP2, Runx2 | Promote hPDLSCs proliferation and osteogenic differentiation | *Lv et al. (2020)* |
| miRNA | PDLSCs | Satb2 | miR-31 promotes Satb2 siRNA and inhibits osteogenic differentiation | Runx2 | Promote osteogenic differentiation of PDLSCs | *Zhen et al. (2017)* |
| miRNA | PDLSCs | Spry1 | Upregulating miR-21 repressing Spry1 and inhibits TNF-α | Spry1; TNF-α | Suppress adipogenic and osteogenic differentiation of PDLSCs | *Yang et al. (2017)* |

Chen et al. (2023), *PeerJ*, DOI 10.7717/peerj.14550

**Table 3** (*continued*)

| Modification | Stem cell | Locus | Pathway mechanism | Target protein | Potential applications | Ref |
|---|---|---|---|---|---|---|
| miRNA | PDLSCs | Notch2 | miR-758 regulated Notch2 and interacts with lncRNA-ANCR | Notch2 | Regulate the osteogenic differentiation of PDLSCs | *Peng et al. (2018)* |
| circRNA | DPSCs | SATB2, RUNX2, OCN | Exosome circLPAR1 induced osteogenic differentiation via downregulation of hsa-miR-31. | SATB2, RUNX2, and OCN | Promote osteogenic differentiation of DPSCs | *Xie et al. (2020)* |
| circRNA | DPSCs | RUNX1, Beclin1 | hsa_circ_0026827 promotes osteoblast differentiation via Beclin1 and the RUNX1 signaling pathways by sponging miR-188-3p | Beclin-1, RUNX1, ALP, OCN and OSX | Promote osteogenic differentiation of DPSCs | *Ji et al. (2020)* |
| circRNA | SCAPs | ALPL | CircSIPA1L1 is sponge for miR-204-5p, which upregulates ALPL. | ALPL | Promote the osteogenic differentiation of SCAPs | *Li et al. (2020b)* |
| circRNA | PDLSCs | SMAD5 | circFAT1 inhibits miR-4781-3p targeting SMAD5. | SMAD5 | Mediate the periodontal bone regeneration of PDLSCs | *Ye et al. (2021)* |
| circRNA | PDLSCs | ERK | CDR1as functioned as an miR-7 sponge to activate the ERK signal pathway. | ERK | Inhibits the proliferation of PDLSCs | *Wang et al. (2019b)* |
| lncRNA | SHEDs | BMP2 | lncRNA C21orf121 competes with BMP2 binding to miR-140-5p, upregulates BMP2 expression. | BMP2, Nestin, $\beta$III-tubulin, MAP2, NSE | Promote neurogenic differentiation of SHEDs | *Liu et al. (2018a)* |
| lncRNA | SCAPs | ALP, RUNX2 | LncRNA-H19 bound to miR-141, elevating phosphorylated levels of p38 and JNK. | SPAG9 | Promote the odontoblast differentiation of SCAPs | *Li et al. (2019b)* |

**Notes.**

DPSCs, dental pulp stem cells; PDLSCs, periodontal ligament stem cells; SHEDs, stem cells from exfoliated deciduous teeth; DFSCs, dental follicle stem cells; SCAPs, stem cells from apical papilla; TLR-4, toll-like receptor 4; Rac1, the Rac family small GTPase 1; RUNX1, runt-related transcription factor 1; CAB39, calcium-binding protein 39; AMPK, AMP-activated protein kinase; mTOR, mammalian target of rapamycin; BMP2, bone morphogenetic proteins 2; MAP2, microtubule-associated protein 2; MAPK8, mitogen-activated protein kinase 8; NSE, neuron-specific enolase; ALPL, alkaline phosphatase; SPARC, secreted protein acidic and rich in cysteine; ZEB2, zinc finger E-box binding homeobox 2; SMAD5, a receptor-regulated SMAD protein in SMAD family member; CDR1as, circRNA CDR1as; TNF-$\alpha$, Tumor necrosis factor-alpha; PHD2, prolyl hydroxylase domain–containing protein 2; LY294402, small interfering RNA for AKT; AKT, a phosphoinositide 3 kinase (PI3K)-dependent serine/threonine.

also be achieved through the binding of specific proteins to sites. Many studies have been conducted on the modification of histones, specifically on methylation and acetylation.

### Histone methylation

Histone methylation refers to the transfer of the methyl group of S-adenosylmethionine (SAM) to arginine or lysine site under the action of histone methyltransferases (HMTs) (*Wang & Jia, 2009*). The expression or repression of genes is associated with specific residues catalyzed by HMTs. For example, histone H3-lysine 4 (H3K4) methylation promotes gene expression, while H3K9 and H3K27 methylation inhibit gene expression (*Blanc & Richard, 2017*). However, histone demethylases can cause histone demethylation. For example, histone demethylase lysine (K)-specific demethylase 1A (KDM1A) targeting H3K4 and H3K9 can affect the differentiation of embryonic stem cells (*Pedersen & Helin, 2010*).

Histone modifications play key roles in the lineage commitment and differentiation of DFPCs and DPSCs. The H3K27me3 mark in DFSCs can strongly suppress the expression of two dentinogenic genes, dentin sialophosphoprotein (DSPP) and dentin matrix protein 1 (DMP1), whereas the H3K27me3 mark is almost absent in the promoters of the genes DSPP and DMP1 in DPSCs and the gene expression levels are significantly higher (*Francis et al., 2020*; *Gopinathan et al., 2013*). The histone methylation-modifying enzyme enhancer of zeste homolog 2 (EZH2) mainly acts on H3K27 and regulates the osteogenic differentiation of DPSCs and DFPCs through the Wnt/$\beta$-catenin signaling pathway. The reduction of EZH2 directly causes the downregulation of H3K27me3 and further leads to the accumulation of $\beta$-catenin, which activates the Wnt/$\beta$-canonical signaling pathway and ultimately promotes the osteogenesis of DPSCs and DFPCs (*Deng et al., 2018b*; *Li et al., 2018a*).

Histone demethylases such as KDM6B, KDM1A, and KDM2A also play a regulatory role in the gene expression of dental-derived stem cells. KDM6B catalyzes the demethylation of histone H3K27me3 located near the promoter of bone morphogenetic protein-2 (BMP2). This activates BMP2 expression and promotes osteogenic and odontogenic growth of dental mesenchymal stem cells (*Hoang et al., 2016*; *Liu et al., 2022*). In addition, KDM6B decreases the level of histone K27 methylation in the promoter of insulin-like growth factor binding protein 5 (IGFBP5), thereby promoting the odontoblast differentiation, proliferation, migration and mineralization of PDLSCs (*Han et al., 2017*).

KDM1A can cooperate with 2-oxoglutarate 5-dioxygenase 2 (PLOD2) to regulate the differentiation process of SACPs (*Wang et al., 2018a*). The knockdown of KDM1A or PLOD2 reduces ALP activity, promotes the expression of DSPP, DMP1 and RUNX2, and enhances bone/dentin production in SCAPs (*Wang et al., 2018a*). Homeobox C8 (homeobox, HOXC8) significantly inhibits the osteogenic differentiation ability of SCAPs by directly binding to the KDM1A promoter and enhancing its transcription (*Yang et al., 2020b*).

KDM2A is able to increase histone H3 lysine 4 (H3K4) trimethylation at the p15$^{INK4B}$ (cyclin-dependent kinase inhibitor 2B) and p27$^{Kip1}$ (cyclin-dependent kinase inhibitor 1B) loci (*Gao et al., 2013*). On the other hand, the attenuation of KDM2A prevents cell cycle

progression in the G1/S phase of SCAPs (*Gao et al., 2013*). Inflammation and hypoxia can also cause the upregulation of KDM2A expression and repress the secreted frizzled-related protein 2 (SFRP2) transcription by reducing histone methylation in the SFRP2 promoter (*Yang et al., 2020a*). SFRP2 can inhibit the Wnt/$\beta$-catenin signaling pathway and further inhibit the target genes of the nuclear factor kappa B (NF-kB) signaling pathway. This enhances the bone/odontogenic differentiation capacity of SCAPs (*Yang et al., 2020a*). Similarly, histone demethylase KDM3B is also capable of regulating the bone/dental differentiation, cell proliferation, and migratory potential of SCAPs (*Zhang et al., 2020*).

### Histone acetylation

Histone acetylation is mainly related to histone acetyltransferases (HATs) and histone deacetylases (HDACs). Under the catalysis of HDACs, the acetyl group of acetyl-CoA is transferred to the amino acid residues of histone tails and promotes gene transcription (*Galvani & Thiriet, 2015*). In addition, histone deacetylases cause chromatin condensation by deacetylating amino acids in histone tails, thereby repressing gene transcription (*Meier & Brehm, 2014*).

Histone acetylation regulates the stemness and differentiation process of dental-derived stem cells. For instance, histone acetyltransferases such as p300, general control non-arrestin 5 (GCN5), and lysine acetyltransferase 6B (KAT6B, also known as MORF) regulate the stemness or osteogenic differentiation of cells by modifying histones on target genes of DPSCs and PDLSCs. Among them, p300 can regulate the expression of genes DMP-1, DSPP, DSP, NANOG, and SOX2 in different ways. p300 promotes the odontogenic differentiation of DPSCs by catalyzing acetylation and promoting the expression of the histone, H3K9, within the promoter regions of DMP-1, DSPP, and DSP (*Liu et al., 2015b*). Furthermore, MORF and GCN5 are mainly involved in the osteogenic differentiation process of PDLSCs under inflammatory conditions, among which GCN5 regulates DKK1 expression through the acetylation of the H3K9 and H3K14 promoter regions (*Li et al., 2016*). DKK1 can inhibit the Wnt/$\beta$-catenin pathway and promote the osteogenic differentiation of PDLSCs. Chronic periodontal inflammation reduces the expression of MORF in PDLSCs (*Xue et al., 2016*). Methoxy-parvacrol (osthole) upregulates MORF in PDLSCs and catalyzes the acetylation of H3K9 and H3K14, which promotes the osteogenic differentiation of PDLSCs under inflammatory conditions (*Sun et al., 2017b*).

In addition, silencing HDAC expression or using histone deacetylase inhibitor (HDACi) can regulate gene expression by inhibiting HDAC activity. The inhibition of HDAC1, HDAC3, and HDAC6 expression can contribute to the odontogenic differentiation of DPSCs. The HDAC inhibitor MS-275 can act on HDAC1 and HDAC3, causing the up-regulation of the gene expression of odontogenic differentiation-related proteins in DPSC, including RUNX2, DMP1, ALP, and DSPP (*Lee et al., 2020*). Similarly, silencing HDAC6 can induce the expression of odontogenic marker genes such as OSX, OCN, and OPN in DPSCs, while inhibiting osteoclast differentiation (*Wang et al., 2018c*). In addition, HDAC6 is also involved in the development and differentiation of PDLSCs. For instance, HDAC6 participates in the aging process of PDLSCs by regulating the acetylation of p27$^{Kip1}$. The inhibition of HDAC6 promotes senescence in PDLSCs and attenuates their osteogenic

differentiation and migration abilities (*Li et al., 2017*). HDAC9, which is mainly involved in the osteogenic differentiation of PDLSCs under inflammatory conditions, impairs the osteogenic differentiation capacity of PDLSCs, whereas miR-17 induces osteogenic differentiation by inhibiting HDAC9 (*Li et al., 2018b*). Finally, HDACi can regulate the differentiation process of dental-derived stem cells by inhibiting HDAC. *Luo et al. (2018)* found that HDACi, trichostatin A, and valproic acid could enhance the acetylation of histones H3 and H4, to promote the proliferation, migration, and adhesion of DPSCs.

## Noncoding RNA

Non-coding RNAs (ncRNA) are transcripts with no or low coding potential, including ribosomal RNA (rRNA), transfer RNA (tRNA), and microRNA (miRNA) (*Ren & Wang, 2021*). miRNA is the most studied in the field of epigenetics (*Ren & Wang, 2021*). miRNAs directly interact with partially complementary target sites located in the 3′ untranslated region of target mRNAs and repress their expression (*Hombach & Kretz, 2016*). Endogenous competing RNAs (ceRNAs) mainly regulate gene expression by competitively binding to miRNA (*Qi et al., 2015*). ceRNAs typically include long noncoding RNAs (lncRNAs) and circular RNAs (circRNAs).

### *miRNA*

miRNAs regulate dental-derived stem cells by affecting the expression of related genes in RUNX2, BMP, Wnt, MAPK, and Notch1 signaling pathways. miRNAs regulate the differentiation process of SCAPs, DPSCs, and PDLSCs by affecting the expression of the gene RUNX2, which mainly mediates osteogenic/odontogenic differentiation in stem cells (*Hussain, Tebyaniyan & Khayatan, 2022*). miR-450a-5p and miR-28-5p can affect the expression of signal transducer and activator of transcription 1 (STAT1), which is mainly involved in the negative regulation of RUNX2 (*Dernowsek et al., 2017*). An *in vitro* model system study found that STAT1 mRNA was gradually down-regulated and RUNX2 mRNA was gradually up-regulated as SHEDs differentiated into osteoblasts (*Dernowsek et al., 2017*). Similarly, miR-218 regulated the mineralization and differentiation process of DPSCs through the ERK1/2 pathway. ERK1/2 signaling converge at Runx2 to control the differentiation of DPSCs (*Chang et al., 2019*).

The transforming growth factor beta (TGF-$\beta$)/ BMP signaling pathway plays an important role in the odontogenic/osteogenic differentiation of dental-derived stem cells. miR-132 inhibits the growth differentiation factor 5 (GDF5) of the TGF-$\beta$ family and activates the NF-$\kappa$B axis, which attenuates the osteogenic differentiation ability of PDLSCs (*Xu et al., 2019*). CD105 is a co-receptor for the type I transmembrane glycoprotein and TGF $\beta$-1, which is associated with the osteogenic differentiation of cells. *Ishiy et al. (2018)* compared the mineralization degree of the SHED matrix with the low/high expression of CD105 and found that the high expression of CD105 reduced osteogenic potential, while miR-1287 was negatively correlated with CD105. miRNA can affect cell differentiation by regulating the expression of the Smad gene, which is an essential transcription factor in the TGF-$\beta$/BMP signaling pathway. miR-135b can inhibit the expression of the Smad4 and Smad5 genes, which hinder the odontoblast-like differentiation of dental pulp cells

(*Song et al., 2017*). In PDLSCs, miR-23a acts on the bone morphogenetic protein receptor type 1B (BMPR1B) gene and inhibits the phosphorylation of Smad1/5/9, which attenuates the osteogenic differentiation of PDLSCs (*Zhang et al., 2019b*). The Smad ubiquitination regulator (Smurf) regulates TGF-$\beta$/BMP signaling through ubiquitination, causing the degradation of signaling molecules and preventing the overactivation of TGF-$\beta$/BMP signaling (*Kushioka et al., 2020*). In SCAPs, miR-497-5p promotes bone/odontogenic differentiation by targeting SMAD-specific E3 ubiquitin protein ligase 2 (Smurf2) and regulating the Smad signaling pathway (*Liu et al., 2020b*). Furthermore, the expression of miR-26a can be upregulated in the exosomes secreted from SHED, and miR-26a can improve angiogenesis in SHED by regulating TGF-$\beta$/SMAD2/3 signaling (*Wu et al., 2021*).

Wnt/$\beta$-catenin signaling can regulate the proliferation, development, and cell fate aspects of dental-derived stem cells. The overexpression of miR-140-5p represses the Wnt1 gene, which affects Wnt/$\beta$-catenin signaling and ultimately inhibits the odontoblast differentiation of DPSCs (*Lu et al., 2019*). Chromodomain helicase DNA-binding protein 8 (CHD8) plays an essential role in maintaining the active transcription of nerve-specific genes and can be targeted and regulated by miR-221 (*Wen et al., 2020*). For example, in SHED, upregulated miR-221 activates the Wnt/$\beta$-catenin pathway by inhibiting CHD8, which promotes the neurogenic differentiation of cells (*Wen et al., 2020*).

In addition, both p38-mitogen-activated protein kinase (MAPK) and neurogenic locus Notch homolog 1 (Notch1) signaling pathways are involved in the osteogenic/odontogenic differentiation process of dental-derived stem cells. miR-143-5p can regulate the expression of MAPK pathway-related genes in DPSCs. To be specific, the downregulation of miR-143-5p increased the expression of p38 MAPK signaling pathway-related genes such as MAPK14 and MKK3/6, and odontoblast differentiation markers such as ALP and OCN (*Wang et al., 2019a*). IGF-I can enhance the odontogenic/osteogenic differentiation ability of mesenchymal stem cells (MSCs) by activating the MAPK pathway, while the IGFBPs/IGF-I complex is regulated by matrix metalloproteinase 1(MMP1) (*Wang et al., 2018b*). In SCAPs, miRNA let-7b inhibits bone/odontogenic differentiation of SCAP by targeting MMP1 (*Wang et al., 2018b*). Notch1 is a transmembrane receptor, and the downregulation of Notch signaling inhibits self-renewal of DPSCs and induces their differentiation (*Wang et al., 2011*). miR-146a-5p can inhibit the expression of Notch1 and regulate the osteogenic/odontogenic differentiation process of DPSCs (*Qiu et al., 2019*).

### ceRNA

*lncRNA.* lncRNA can regulate the differentiation of dental-derived stem cells by directly acting on GDF5, distal-less homeobox 3 (DLX3), and Kruppel-like factor 2 (KLF2). lncRNA growth arrest specific transcript 5 (GAS5) can enhance the expression of GDF5 in cells and promote the phosphorylation of the p38 MAPK/JNK signaling pathway, which enhances the osteogenic differentiation of PDLSCs (*Yang et al., 2020c*). lncRNA H19 inhibits DNMT3B-mediated methylation of the DLX3 gene through S-adenosyl-L-homocysteine hydrolase (SAHH), which regulates odontoblast differentiation of DPSCs (*Zeng et al., 2018a*). The direct interaction of lncRNA SNHG1 with EZH2 regulates KLF2 promoter H3K27me3

methylation and inhibits the differentiation of PDLSCs to osteoblasts (*Li, Guo & Wu, 2020c*).

By inhibiting the expression of miRNAs, lncRNAs can also play a regulatory role. During the osteogenic differentiation of PDLSCs, lncRNAs can act as ceRNAs and form networks to regulate the Wnt/$\beta$-catenin signaling pathway (*Lai et al., 2022*). lncRNA-ANCR competitively binds miR-758 and inhibits the expression of Notch2, which further affects the Wnt/$\beta$-catenin signaling pathway and inhibits the osteogenic differentiation of PDLSCs (*Peng et al., 2018*). FoxO1 promotes bone formation in PDLSCs by competing with TCF-4 for $\beta$-catenin and inhibiting the Wnt pathway (*Wang et al., 2016*). lncRNA-POIR can inhibit the expression of the miR-182 target gene FoxO1 and affect the osteogenic differentiation process of PDLSCs (*Wang et al., 2016*). As ceRNAs, lncRNAs can affect the expression of genes related to the MAPK and BMP signaling pathways. LncRNA-H19 can competitively bind to miR-141 and prevent the miRNA-mediated degradation of SPAG9, thereby increasing the phosphorylation levels of p38 and JNK, which promotes the bone/odontogenic differentiation of SCAPs (*Li et al., 2019b*). In SHEDs, lncRNA C21 or f121 can compete with BMP2 to bind with miR-140-5p and promote the neurogenic differentiation of SHEDs by upregulating BMP2 expression (*Liu et al., 2018a*). lncRNA-CCAT1 combined with miR-218, and lncRNA G043225 combined with miR-588 can promote the odontogenic differentiation of DPSCs (*Chen et al., 2020a*; *Zhong et al., 2019*).

## circRNA

In PDLSCs, circRNAs can indirectly regulate osteogenic differentiation by binding to miRNAs (*Gu et al., 2017*). circRNA cerebellar degeneration-related protein 1 transcript (CDR1as) and miR-7 can regulate the osteogenic differentiation and stemness of PDLSCs. CDR1as may promote the upregulation of GDF5 and the phosphorylation of Smad1/5/8 and p38 MAPK by inhibiting the expression of miR-7, inducing the differentiation of PDLSCs to osteoblasts. In addition, the interaction of CDR1as with miR-7 can also upregulate the expression of KLF4 to maintain the stemness of PDLSCs, while RNA-binding protein hnRNPM regulates its expression in PDLSCs by interacting with CDR1as (*Gu et al., 2021*). During the osteogenic differentiation of SCAPs, the expression profiles of circRNAs are significantly altered, and circRNAs mainly function as ceRNAs (*Li et al., 2019a*). circ SIPA1L1 can promote the expression of the gene ALPL (alkaline phosphatase alkaline phosphatase) by binding to miR-204-5p, which causes the osteogenic differentiation of SCAPs (*Li et al., 2020b*).

## Epigenetic regulatory network

In the epigenetic regulation of dental-derived stem cells, there are multiple links between histone modifications, DNA methylation, and ncRNA, which interact with each other and participate in genetic regulation together. ncRNAs participate in the regulation of gene expression in stem cells by regulating DNA methylation. For example, lncRNA H19 can inhibit the activity of DNMT3B, which reduces the methylation of the distal-less homeobox (DLX3) of the gene, thereby promoting the odontogenic differentiation of DPSCs (*Zeng et al., 2018a*). Similarly, miR-675 can also promote the odontogenic differentiation of

DPSCs by inhibiting DNMT3B (*Zeng et al., 2018b*). The lncRNA HOTAIRM1 inhibits the expression and enrichment of DNMT1 on the HOXA2 promoter and mechanically binds to the CpG island in the HOXA2 promoter region, leading to hypomethylation and the induction of HOXA2 and DFSC differentiation into osteoblasts (*Chen et al., 2020b*).

ncRNAs can also play a role in histone modifications, including histone methylation, histone acetylation, and histone deacetylation. miR-153-3p inhibits the transcription of ALP, Runx2, and OPN by targeting KDM6A, which results in the attenuated osteogenic differentiation of PDLSCs (*Jiang & Jia, 2021*). miRNAs are involved in the aging and differentiation process of dental-derived stem cells by regulating the expression of HAT or HDAC. The upregulation of miR-152 represses HAT sirtuinc 7 (SIRT7) expression and affects the degree of histone acetylation, which accelerates the aging process of DPSCs (*Gu et al., 2016*). The upregulation of miRNA-383-5p can promote the down-regulation of the HDAC9 mRNA level, which leads to increased alkaline phosphatase activity, mineral node formation, and the expressions of RUNX2, osteocalcin, and Smad4 in PDLSCs and other osteogenic markers (*Ma & Wu, 2021*). Similarly, miR-22 can inhibit HDAC6 expression and promote the osteogenic differentiation of PDLSCs (*Yan et al., 2017*).

## Therapeutic application of dental-derived stem cells in dental pulp and periodontal regeneration

In 1971, *Nygaard-Ostby & Hjortdal (1971)* proposed the concept of pulp tissue regeneration. Pulp regeneration refers to the formation of new pulp tissue through tissue engineering to replace the infected or necrotic pulp tissue, thereby restoring the structure and function of the pulp-dentin complex under physiological conditions. Conventional apexogenesis may result in the thinning of the dentin wall and underdevelopment of the root, which greatly increases the risk of long-term root fracture. However, pulp regeneration can effectively form healthy pulp tissue and promote the formation of dentin. Periodontitis, a chronic inflammation of the periodontal tissue caused by dental plaque, can cause the destruction and absorption of the alveolar bone and tooth loss. Traditional periodontal treatments, such as scaling, focuses on controlling the occurrence of inflammation, but fails to restore the structure and function of periodontal tissue entirely. Periodontal tissue regeneration reconstructs periodontal tissue damaged by periodontitis and restores its structure and function by means of tissue engineering (*Chen et al., 2010*). The key elements of tissue regeneration are stem cells, scaffolds, and signaling molecules. The biological behavior of stem cells is regulated by epigenetics. Further research will likely lead to the discovery of an increasing number of new factors. These factors may regulate the development of stem cells towards odontogenic differentiation, angiogenesis, neurogenesis, and osteogenic differentiation through epigenetic mechanisms, and may facilitate the application of pulp regeneration and periodontal regeneration. Figure 1 demonstrates the epigenetic regulation of dental-derived stem cell differentiation and its application in pulp regeneration and periodontal regeneration.

### Odontogenic differentiation

Transcription factor RUNX2 mainly mediates osteogenic/odontogenic differentiation and may effectively promote the expression of dentin matrix proteins or induce the

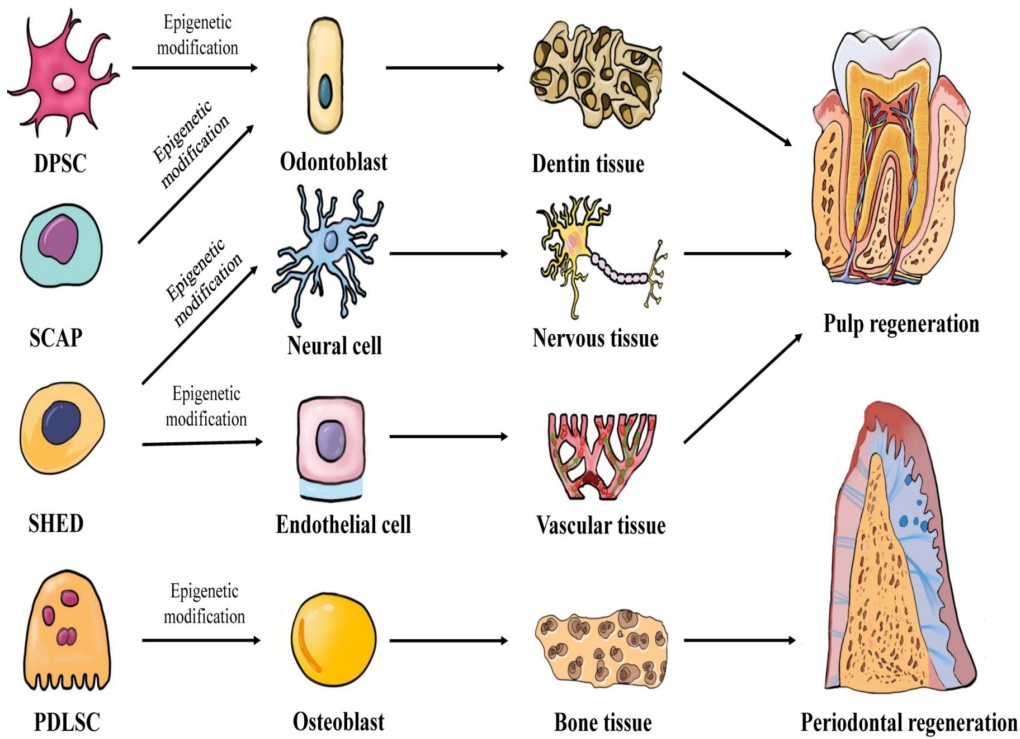

**Figure 1** **Multilineage potential of human dental-derived stem cells.** Four kinds of human dental-derived stem cells have the capacity to differentiate under epigenetic modification into different somatic cell and tissue types, and finally contribute to regeneration of pulp or periodontal tissue. DPSC, Dental pulp stem cell; SCAP, Stem cells from apical papilla; SHED, Stem cells from human exfoliated deciduous teeth; PDLSC, Periodontal ligament stem cell.

transdifferentiation of cells into osteoblasts (*Li et al., 2011*). HDACi can affect the expression of the RUNX2 gene in stem cells by acting on HDAC. MS-275 can inhibit the expression of HDAC1 and HDAC3 and may induce the up-regulation of odontogenic related proteins in DPSCs, including RUNX2, DMP1, ALP, and DSPP, which promotes the odontogenic differentiation of DPSCs (*Lee et al., 2020*). *Sultana et al. (2021)* showed that without the induction of mineralized medium, MS-275 alone could increase the expression levels of BMP2, DMP1, DSPP, and Runx2 mRNA of mouse odontoblast-like cell line MDPC-23, and improved ALP activity. Therefore, MS-275 can effectively promote the odontogenic differentiation of DPSCs.

BMPs signal through canonical Smad and non-Smad signaling pathways, in which BMP-Smad signaling can be involved in the formation of coronal dentin (*Omi et al., 2020*). The HDACi inhibitor TSA can significantly upregulate the levels of Smad and NFI-C in DPSCs by inhibiting HDAC3. *Jin et al. (2013)* treated DPSCs with TSA and found that the expression of BSP, DMP1, and DSPP was significantly increased compared with the control group; the level of Smad2/3 was also significantly up-regulated 21 days after mineralization induction. In contrast, neonatal mice that were maternally exposed to TSA exhibited thicker dentin and more dentin cells in their postpartum molars, with a

greater ability to secrete DSP (*Jin et al., 2013*). In addition to regulating gene expression in DPSCs, *Duncan et al. (2017)* demonstrated that TSA could also promote the release of dentin matrix components from dentin. These two studies show that HDACi can promote the differentiation of DPSCs into odontoblasts as well as the release of the dentin matrix, which is very beneficial to the repair of dental pulp-dentin complex.

The Wnt/$\beta$-catenin signaling pathway may regulate the process of dentin formation and tooth development and the amplification of Wnt signaling can significantly improve the survival rate of damaged dental pulp cells and promote tertiary dentin formation (*Hunter et al., 2015*). miR-140-5p can repress the Wnt1 gene and affect the Wnt/$\beta$-catenin signaling process (*Lu et al., 2019*). *Lu et al. (2019)* collected impacted third molars from patients aged 14–22 years and divided the extracted DPSCs into an miR-140-5p inhibition group, a negative control group (NC), and a blank control group. After 14 days of inducing cells to differentiate into odontoblasts, Alizarin Red S staining showed that the mineralized matrix deposition was greatest in the inhibitor group and least in the mock group. Western blotting showed that the inhibitor group had the highest expressions of DSPP and DMP-1 proteins while the mock group had the lowest (*Lu et al., 2019*). These results indicate that miR-140-5p can affect the odontoblast differentiation process of DPSCs.

The p38 MAPK pathway is central to the transcriptional control of odontoblasts and its activation is critical for apical morphogenesis and enamel secretion (*Greenblatt et al., 2015*). The activation of the MAPK signaling pathway is also associated with the osteogenic/odontogenic differentiation of DPSCs (*Wu et al., 2019*). lncRNA-H19 can competitively bind to miR-141 and upregulate the phosphorylation levels of p38 and JNK. *Li et al. (2019b)* induced transfected SCAPs in an osteoblast differentiation medium, and the Western blot results showed that the protein expressions of OCN, RUNX2, ALP, and DSP in the H19-infected SCAP group were significantly higher than those in the control group. The SCAP that had a stable expression of H19, and the control group, were further loaded on Bio-Oss collagen scaffolds and implanted in the subcutaneous tissue of nude mice. H&E and Masson staining showed that the abundance of bone-like structures, collagen deposition, and dentin-like structures in the H19-infected SCAP group was higher than that in the control group (*Li et al., 2019b*). This indicates that lncRNA-H19 can promote the odontoblast differentiation process of SCAPs by activating the p38 and JNK signaling pathways.

### Nerve regeneration

CHD8 can affect neural progenitor cells and neurons, and also plays a role in maintaining the active transcription of neural-specific genes (*Wilkinson et al., 2015*). Meanwhile, CHD8 can also alter neurogenesis and cortical development by regulating the Wnt/$\beta$-catenin signaling pathway (*Platt et al., 2017*). In SHEDs, upregulated miR-221 can bind to CHD8 and activate the Wnt/$\beta$-catenin pathway (*Wen et al., 2020*). *Wen et al. (2020)* divided the SHEDs in the third-generation logarithmic growth phase into six groups: blank group, NC group (transfected with miR-221 negative sequence), miR-221 mimic group (transfected with miR-221 mimic), miR-221 inhibitor group (transfected with miR-221 inhibitor), siRNA-CHD8 group (transfected with siRNA into CHD8 vector), and miR-221 inhibitor

+ siRNA-CHD8 group (co-transfected with miR-221 inhibitor and siRNA-CHD8). The results of Western blot analysis revealed that the expressions of NSE, NESTIN, MAP-2, NF-M, and TH in the miR-221 inhibitor group were significantly lower than those in the NC group, while the miR-221 mimic group and siRNA-CHD8 group were both lower than those in the NC group. Immunofluorescence examination showed that the expressions of NSE and MAP-2 in the miR-221 inhibitor + siRNA-CHD8 group were higher than that in the miR-221 inhibitor group (*Wen et al., 2020*). Among them, neuron-specific enolase (NSE) was a highly specific marker of neurons and peripheral neuroendocrine cells, NESTIN was a key early neural progenitor cell marker, NF-M and microtubule-associated protein 2 (MAP2) was a neuron-associated marker, and TH was the rate-limiting enzyme in dopamine neurotransmitter biosynthesis. These results suggest that miR-221 can promote SHED differentiation into neurons by inhibiting CHD8.

BMP2 is a neurotrophic factor that induces the growth of brain dopaminergic (DA) neurons *in vitro* and *in vivo*, whose induction depends on the Smad signaling pathway (*Hegarty, Sullivan & O'Keeffe, 2013*). In SHEDs, lncRNA C21 or f121 competitively binds to miR-140-5p and upregulates BMP2 expression (*Liu et al., 2018a*). The results of bioinformatics analysis conducted by *Liu et al. (2018a)* showed that there was a targeting relationship between the second spliceosome of lncRNA C21 or f121 and miR-140-5p, the same as miR-140-5p and BMP2. This suggested that lncRNA C21 or f121 competed with BMP2 to bind to miR-140-5p. *Liu et al. (2018a)* grouped and experimented with SHEDs in the third-generation logarithmic growth phase. The results showed that the protein expressions of both Nestin and $\beta$III-tubulin decreased, but increased in the transfected miR-140-5p inhibitor group compared with the NC group (transfected with lncRNA C21 or f121 negative sequence), the BMP2 and MAP2 in the si-C21 or f121 group, the miR-140-5p group, and the si-C21 or f121+miR-140-5p group. Further experiments showed that the up-regulation of lncRNA C21 or f121 or down-regulation of miR-140-5P increased the frequency of social behavior in rats and decreased the cumulative time of repetitive stereotyped movements in young rats (*Liu et al., 2018a*). All of the abovementioned studies show that lncRNA C21 or f121 can effectively promote the neurogenic differentiation of SHEDs.

Finally, some HDACi have the effect of inducing neurogenic differentiation of cells. For instance, *Okubo et al. (2016)* found that the total number of mRNAs of mature neuronal markers, neurofilament medium polypeptide (NeFM), and microtubule-associated protein 2 (MAP2) significantly increased to approximately 80% in VPA-treated rats compared with untreated rats (*Okubo et al., 2016*). Other studies have shown that the neurite number on the cells increased and branched processes were elongated after treating MSCs with combinations of MS-275 or NaB (a kind of HDACi). The cells were visualized by immunofluorescence staining of the neuronal markers (*Jang et al., 2019*). The above studies demonstrate the role of HDACi in inducing neurogenic differentiation. Further research may reveal whether dental-derived stem cells can be inducted to differentiate into neural cells.

## Angiogenesis

TGF-$\beta$/SMAD2 signaling can promote angiogenesis and the secretion of vascular endothelial growth factor (*Ji et al., 2014*). *Wu et al. (2021)* discovered that the expression of miR-26a was up-regulated in SHED-secreted exosomes (SA-Exo), and miR-26a could promote the expression of TGF-$\beta$/SMAD2/3 signaling. The expression of angiogenesis-related proteins (VEGF, angiopoietin 2 and PDGF) of SHEDs was up-regulated, and the endothelial differentiation potential was increased after being treated with SA-Exo. SA-Exo treatment also increased the expression levels of angiogenesis-related proteins in HUVECs. *Wu et al. (2021)* implanted SHED aggregates into immunodeficient mice and performed histological analysis, which revealed the formation of a new, continuous dentin layer and blood vessels, and the regeneration of the dentin-pulp complex. In addition, dentin and blood vessel formation were enhanced by the combined implantation of SHED aggregates and SA-Exo. The expression level of the angiogenic marker CD31 was also higher. The inhibition of SA-Exo repressed dentin-pulp complex regeneration; however, supplementation with exogenous SA-Exo could restores this process. The results of qRT-PCR confirmed that the expression of miR-26a was significantly increased in SA-Exo, and the inhibition of miR-26a in SA-Exo could not cause the endothelial differentiation of SHED and HUVECs. Western blot analysis revealed that the overexpression of miR-26a upregulates TGF-$\beta$/SMAD2/3 signaling, and the inhibition of these two pathways led to reduced endothelial differentiation in SHEDs and HUVECs (*Wu et al., 2021*). These studies confirm that miR-26a in SA-Exo promote angiogenesis in SHEDs through the TGF-$\beta$/SMAD2/3 signaling pathway.

ncRNAs have a strong ability to regulate endothelial cell migration, proliferation, and differentiation. miR-30a-3p targets the epigenetic factor methyl-CpG-binding protein 2 (MeCP2), and the overexpression of MeCP2 damages important genes involved in the regulation of endothelial function such as sirtuin1 (*Volkmann et al., 2013*). *Volkmann et al. (2013)* transfected endothelial cells with miR-30a-3p precursors, which significantly reduced MeCP2 protein levels and increased the migratory ability of endothelial cells. This suggests that miR-30a-3p has the ability to regulate endothelial cells. In addition, lncRNAs also have a role in regulating endothelial cells. *Neumann et al. (2018)* discovered that lncRNA GATA6 could inhibit the action of the epigenetic regulator, LOXL2, reduce the endothelial-mesenchymal transition *in vitro*, and promote the formation of blood vessels in mice. These two studies demonstrated the potential of ncRNAs in promoting angiogenesis. Future studies may reveal whether they can regulate dental-derived stem cells for angiogenesis.

## Osteogenic differentiation

Different dental-derived stem cells have a diverse range of DNA methylation levels and unique osteogenic differentiation potentials. Compared with DPSCs and DFPCs, PDLSCs have lower methylation levels of genes related to osteogenesis, higher expression levels of factors such as SMAD3, ALP, OCN, and RUNX2, and a higher osteogenic differentiation potential (*Ai et al., 2018*). After culturing PDLSCs, DFPCs, and DPSCs14 in osteoinductive medium, *Ai et al. (2018)* used Alizarin Red S positive staining and found that the relative

intensity of staining in PDLSCs was significantly higher than that in DPFCs and DPSCs. The simultaneous subcutaneous transplantation of cell deposits mixed with hydroxyapatite onto the dorsal surface of immunocompromised male mice found that PDLSCs formed more osteoid. This study demonstrated that DNA methylation can regulate the osteogenic differentiation potential of dental-derived stem cells by affecting the expression of related genes.

Insulin-like growth factor (IGF) and its binding proteins play an important role in promoting bone formation (*Nguyen et al., 2013*). KDM6B can promote IGFBP5 transcription by reducing histone K27 methylation (*Han et al., 2017*). *Han et al. (2017)* found that by administering a local injection of rhIGFBP5 into the periodontitis area of a piglet model, they could significantly promote the regeneration of periodontal tissues such as alveolar bone and gingiva after 12 weeks.

The Wnt/$\beta$-catenin pathway can promote/inhibit the osteogenic differentiation of cells under various conditions (*Wagner et al., 2011*). HAT GCN5 inhibits the Wnt/$\beta$-catenin signaling pathway by increasing the levels of H3K9ac and H3K14ac in the DKK1 promoter region (*Li et al., 2016*). *Li et al. (2016)* found that more active osteogenic differentiation was presented in cell populations with higher GCN5 expression, and GCN5 downregulation may lead to defective osteogenic differentiation of PDLSCs. GCN5 knockdown resulted in increased expression of $\beta$-catenin and decreased expression of genes and proteins related to osteogenic differentiation, such as RUNX2 and ALP. ChIP assays indicated that GCN5 binds to the promoter region of DKK1. Alveolar bone loss in the first and second maxillary molars was significantly reduced with increased GCN5 expression in periodontital rats (*Li et al., 2016*). Therefore, HAT GCN5 can promote the osteogenic differentiation of PDLSCs to regenerate alveolar bone by inhibiting the Wnt/$\beta$-catenin signaling pathway.

Both the MAPK and TGF-$\beta$/Smad signaling pathways may be involved in BMP-mediated osteogenesis (*Kim, Park & Choung, 2018*; *Zhu et al., 2018*). CDR1as is an inhibitor of miR-7 that can cause the upregulation of TGF-$\beta$ family member GDF5, and the phosphorylation of p38 MAPK (*Li et al., 2018c*). In PDLSCs, the knockdown of CDR1as or the overexpression of miR-7 significantly suppressed the mRNA and protein levels of GDF5. In contrast, a lower expression of GDF5 resulted in a decrease in the osteogenic markers ALP and RUNX2, as well as phosphorylated p38 MAPK. *Li et al. (2018c)* loaded CDR1as siRNA and negative control siRNA-treated PDLSCs onto scaffold material and implanted this into the calvarial defect area of nude mice. The results showed that the CDR1as knockdown group had less bone formation and significantly lower new bone formation than the control group. In the control group, bone tissue was generated at the edges of the bone defects, but in the CDR1as knockout group, little new bone was observed (*Li et al., 2018c*). This study strongly demonstrates that CDR1as can promote the osteogenic differentiation of PDLSCs.

## CONCLUSIONS

Dental-derived stem cells, as mesenchymal stem cells, play an essential role in pulp and periodontal regeneration. Epigenetic regulation can adjust gene expression independent

of DNA sequence changes, which can affect the proliferation, differentiation, and function of dental-derived stem cells. DNA methylation, histone modifications, and ncRNAs constitute a grand epigenetic regulatory network that can function independently or coherently. Pulp regeneration and periodontal regeneration can be well achieved through epigenetic regulation.

Considering the various types of epigenetic modifications and different mechanisms, the epigenetic research on dental-derived stem cells is still lacking at this stage. Current research has focused on classical epigenetic modifications and modification sites, while some potential modifications such as DNA 6mA modification, mRNA m6A modification, and modification on tRNA still need more experimental verifications. Epigenetic modifications are widespread in eukaryotes; some modification mechanisms in mesenchymal stem cells can be further studied in dental-derived stem cells. As new functions of epigenetic modification are revealed, we can also focus on their regulatory roles in dental-derived stem cells.

Current research has focused on the regulatory role of specific epigenetic modification mechanisms in dental-derived stem cells. However, less attention has been paid as to whether there are interactions between epigenetic modifications, which limits our further exploration of epigenetic regulatory networks.

Studies on the epigenetic regulation of dental-derived stem cells have also been influenced by the cells themselves. The special odontogenic potential of DPSCs and the excellent osteogenic potential of PDLSCs make them the best choice for pulp regeneration and periodontal regeneration. Therefore, the epigenetic mechanisms of DPSCs and PDLSCs are currently the most studied. SCAPs and SHEDs, which are obtained from tooth roots and exfoliated deciduous teeth respectively, have also been widely studied due to their abundant sources and low immunogenicity. In contrast, the sources of DFPCs are more limited, leading to fewer studies.

Finally, the application of the epigenetic regulation of dental-derived stem cells is influenced by its mechanism of action. Tissue engineering can be accomplished through stem cells, scaffolds and signaling molecules, all of which are applicable. However, epigenetic regulation needs to act on specific modification sites, and most of the current experiments are done in the form of virus transfection, greatly limiting the application of epigenetic regulation in tissue regeneration. In addition, dental-derived stem cells interfere with the proliferation and differentiation of surrounding cells by secreting exosomes. Therefore, further research on exosomes will be beneficial to the application of epigenetic regulation in dental-derived stem cells.

### Funding

This work was supported by the National Natural Science Foundation of China (No. 81800929 and No. 82170921), the Chengdu Science and Technology Program (No.2019-YF05-00441-SN) and the Research and Develop Program, West China Hospital of

Stomatology Sichuan University (No. LCYJ2019-24). The funders had no role in study design, data collection and analysis, decision to publish, or preparation of the manuscript.

### Grant Disclosures

The following grant information was disclosed by the authors:

National Natural Science Foundation of China: 81800929, 82170921.

Chengdu Science and Technology Program: 2019-YF05-00441-SN.

Research and Develop Program, West China Hospital of Stomatology Sichuan University: LCYJ2019-24.

### Competing Interests

The authors declare there are no competing interests.

### Author Contributions

- Yuyang Chen conceived and designed the experiments, performed the experiments, analyzed the data, prepared figures and/or tables, authored or reviewed drafts of the article, and approved the final draft.
- Xiayi Wang performed the experiments, analyzed the data, prepared figures and/or tables, authored or reviewed drafts of the article, and approved the final draft.
- Zhuoxuan Wu performed the experiments, analyzed the data, prepared figures and/or tables, and approved the final draft.
- Shiyu Jia analyzed the data, prepared figures and/or tables, and approved the final draft.
- Mian Wan conceived and designed the experiments, analyzed the data, authored or reviewed drafts of the article, and approved the final draft.

### Data Availability

This is a literature review and there is no raw data.

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
