# Peer review of "Epigenetic regulation of dental-derived stem cells and their application in pulp and periodontal regeneration"

_PeerJ, doi:10.7717/peerj.14550_

## Round 0.1 · original submission · Major Revisions

· Academic Editor

Major Revisions

The current review article has gained a lot of interest in recent years and the submitted manuscript covers this extensively. However, there are some comments that need to be addressed and are provided by the reviewers. Please address the comments and make the necessary changes before resubmitting the manuscript. Additionally, from my review of the manuscript, I recommend that you recheck for grammatical and typographical errors.

·

Basic reporting

Dear,
This is an interesting article but you just repeat those data that have published in REVIEW articles, you did not collect the the newly original studies, I suggest you to make table and collect the new studies (2010-2022).

Experimental design

It would be better to remove the many repeated subtitles in published articles such as 4.1DNA methylation, 4.2 Histone modifications,......

Validity of the findings

Author should explained and discussed more about recent studies (2015-2022)

Reviewer 2 ·

Basic reporting

The paper is a good review of studies that focused on creating epigenetic modifications to dental-derived stem cells for application in pulp and periodontal regeneration.

Experimental design

The tables are very self-explanatory and give a broad overview of all the studies in the field.

Validity of the findings

No comment

Additional comments

In line 430, the authors mention a word called “apexoplasty” which is an undefined word. I would request the authors to look into it.

Reviewer 3 ·

Basic reporting

I checked the " Epigenetic regulation of dental-derived stem cells and their application in pulp and periodontal regeneration" file.
The article "The Role of Epigenetic in Dental and Oral Regenerative Medicine by Different Types of Dental Stem Cells: A Comprehensive Overview" thoroughly reviewed the different aspects of epigenetics in dental regenerative medicine and even analyzed the existing documents related to different types of dental stem cells. Also, this article's structure is almost identical to the mentioned reference.
The next point is that this article has multiple grammatical and typographical mistakes.

Experimental design

no comment

Validity of the findings

no comment

---

## Round 0.2 · accepted · Accept

· Academic Editor

Accept

The authors have addressed all the reviewers' comments to satisfaction and both the reviewers have endorsed the article to be accepted.

·

Basic reporting

Dear,
It is acceptable for publication,

Experimental design

Dear,
It is acceptable for publication,

Validity of the findings

Dear,
It is acceptable for publication,

Additional comments

Dear,
It is acceptable for publication,

Reviewer 2 ·

Basic reporting

No comment

Experimental design

No comment

Validity of the findings

No comment

Additional comments

After thoroughly going through the revised manuscript, I am satisfied with all the corrections the authors have made after incorporating the points raised by the reviewers